# Translation, Cross-Cultural Adaptation, and Psychometric Properties of Writing Readiness Inventory Tool in Context (WRITIC)

**DOI:** 10.3390/children10030559

**Published:** 2023-03-16

**Authors:** Paulo Delgado, Filipe Melo, Liesbeth de Vries, Margo Hartingsveldt, Ana Rita Matias

**Affiliations:** 1Comprehensive Health Research Centre (CHRC), Department of Sport and Health, School of Health and Human Development, University of Évora, 7004-516 Évora, Portugal; paulolourencodelgado@gmail.com; 2Comprehensive Health Research Centre (CHRC), Faculty of Human Kinetics, University of Lisbon, 1649-004 Lisbon, Portugal; fmelo@fmh.ulisboa.pt; 3Centre of Expertise Urban Vitality, Faculty of Health, Amsterdam University of Applied Sciences, 1000 CC Amsterdam, The Netherlands; liesbeth@bee-kinderergotherapie.nl; 4Occupational Therapy Programme, Hanze University of Applied Sciences, 9714 CA Groningen, The Netherlands

**Keywords:** handwriting, preschool, psychometric properties, WRITIC-PT-TP

## Abstract

This article describes the translation and cultural adaptation process of the WRITIC (Writing Readiness Inventory Tool in Context) into European Portuguese. We examined the content and convergent validity, test-retest, and interrater reliability on the norm-referenced subdomain of the Portuguese (PT) WRITIC Task Performance (TP). To establish content validity, we consulted six experts in handwriting. Internal consistency was found with 70 children, test-retest reliability with 65, inter-rater reliability with 69, and convergent validity with 87. All participants were typically developing kindergarten children. Convergent validity was examined with the Beery–Buktenica Developmental Test of Visual-Motor Integration (Beery™VMI-6) and the Nine Hole Peg-Test (9-HPT). On content validity, we found an agreement of 93%, a good internal consistency with Cronbach’s alpha of 0.72, and an excellent test-retest and inter-rater reliability with ICCs of 0.88 and 0.93. Correlations with Beery™VMI-6 and 9-HPT were moderate (r from 0.39 to 0.65). Translation and cross-cultural adaptation of WRITIC into European Portuguese was successful. WRITIC-PT-TP is stable over time and between raters; it has excellent internal consistency and moderate correlations with Beery™VMI-6 and 9-HPT. This analysis of the European Portuguese version of WRITIC gives us the confidence to start the implementation process of WRITIC-PT in Portugal.

## 1. Introduction

Handwriting is a complex task that depends on different skills, such as cognitive, motor, perceptual, and sensory [1], closely related to composition and literacy outcomes [2]. In our digital age, with increasing technology use in childhood, research in first-grade children showed a decrease in overall motor skills and an increase in static pencil grip use compared to twenty years ago [3]. Despite the increasing use of computers, tablets, and smartphones (where children learn typing), handwriting remains an essential skill for children to participate in school [4]. Handwriting is essential for learning reading and spelling. The brain has close functional relationships between the reading and writing processes [2]. James and Engelhart showed that handwriting experience influences brain activation during letter perception. For these authors, handwriting is important for the early conscription in letter processing of brain regions which are the basis of successful reading and may facilitate its achievement in young children [5]. Research by Ose Askvik et al. stated that it is vital that handwriting and drawing should be kept in a learning environment. This is because of the benefits of sensory-motor integration due to the larger involvement of the senses as well as fine and precisely controlled hand movements when writing by hand and when drawing [6]. Research in the United States shows that the time spent on handwriting has decreased in recent years [7]. Consequently, less time spent on handwriting means less exercise, and it becomes more difficult to automize this process. When handwriting is not automated, working memory is continuously needed, and fewer resources are available for cognition and text composition, affecting academic performance and self-esteem [8]. However, a lot is still handwritten in primary school. Research in Australia and the US shows that kindergarteners are still busy with paper-and-pencil tasks for about an hour a day [9]. 

Two performance components have been identified that must be mastered before a child can succeed with handwriting [10]: fine motor coordination [11] and visual motor integration [12]. Fine motor coordination can be tested by the Nine-Hole Peg Test (9-HPT) [13], and visual-motor integration can be tested by the Beery–Buktenica Developmental Test of Visual-Motor Integration (Beery™VMI-6) [14].

In Portugal, where handwriting is the first modality taught and has been dominant throughout the years [15], a general framework of methodological guidelines is provided that support children’s active engagement in handwriting activities, with an emphasis on the communicative nature of handwriting [16]. Portuguese kindergartners learn prewriting skills at the age of 5 to 6 years. During this period, children learn an appropriate sitting posture, to handle the pencil properly, and to produce different writing patterns before they start handwriting with cursive or block letters in first grade. In Portugal, children enter first grade normally in the school year in which they turn six before 1 January.

Difficulties in paper-and-pencil tasks often arise early, so analyzing these skills before entering school (grade 1) is critical to ensuring that teachers know what to do and how to act in children who are not ready to start the handwriting learning process [17].

Handwriting readiness is a topic that needs to be deepened in early childhood literature. However, it is known that children with handwriting difficulties develop negative experiences in this area, including frustration, self-efficacy, and low motivation [18]. Identifying kindergarten children at risk of developing handwriting difficulties may allow early intervention to prevent handwriting difficulties and negative secondary effects in later grades. 

Depending on the country and the dysgraphia definition, difficulties range between 10% and 30% [19]. Many studies mentioned that children with handwriting difficulties have deficits in fine motor coordination [1,20]. It also mentioned the correlation between visual motor integration and poor handwriting [21,22,23]. Volman et al. assumed that fine motor coordination and visual-motor integration are two different performance components in children’s handwriting development [14].

WRITIC assessment was developed because no known instrument assesses all components of handwriting readiness based on Berninger’s conceptual model [24,25]. WRITIC-assessment evaluates handwriting readiness in the classroom context and evaluates the child’s interest in paper and pencil tasks, the physical and social environment, and the performance of age-appropriate paper-and-pencil tasks, such as coloring, writing patterns, writing their own name, and copying letters and numbers. WRITIC assessment discriminates between children who are ready for instruction in handwriting and children who are not [24]. Early identification of possible difficulties in handwriting readiness is the best way to act if necessary [25]. If we can prepare children for the proper use of handwriting tools in their paper-and-pencil tasks, less energy and attention are needed to achieve success. Early intervention is the principle to achieve this goal. 

Early identification is possible with a WRITIC assessment. Given the need and relevance of translating an instrument of this nature and with these characteristics for the Portuguese population, the translation and cross-cultural adaptation were made. When performed well, translation and adaptation of existing assessments improve the accessibility of construct estimation to a broader population, thus facilitating greater use in applied research and counseling practice [26]. 

Translation and cross-cultural adaptation are processes that involve the assessment of conceptual, semantic, and content equivalences of the constructs [27] to assure cross-cultural representativeness, comparability of scores, and reliability and validity, also minimizing confounds such as construct irrelevance [28,29]. Consequently, it is important to study reliability and validity in the target Portuguese population.

Therefore, this study aims to examine the content validity, internal consistency, test-retest, interrater reliability, and convergent validity with Beery™VMI-6 and 9-HPT of WRITIC with Portuguese kindergarten children of five to six years old.

## 2. Materials and Methods

The original WRITIC was cross-culturally adapted from English to Portuguese. Then, pilot testing was carried out with 88 children of five to six years old, following up the assessment of the psychometric properties of the Portuguese version (WRITIC-PT) for (a) content validity, (b) reliability, and (c) convergent validity.

### 2.1. Ethics Approval

The Évora University research ethics committee approved all the procedures. In addition, the participants’ parents gave written informed consent, according to the Helsinki declaration, before participating in this study. The children provided assent.

### 2.2. WRITIC

This instrument allows us to assess handwriting readiness in children aged 5 and 6 and aims to identify whether children are ready to learn the skill of handwriting. WRITIC has items in three domains and six subdomains: Child (Interest and Sustained attention), Environment (Physical and Social), and Tasks (Performance and Intensity).

WRITIC is administered in the classroom, where the influence of the context is considered. The WRITIC contains items of three domains: ‘Child’, ‘Environment’, and ‘Paper-and-pencil tasks’. Every domain consists of two subdomains: the ‘Child’ domain includes ‘Interest’ and ‘Sustained attention’, the ‘Environment’ domain includes ‘Physical environment’ and ‘Social environment’, and the paper-and-pencil tasks domain includes ‘Task performance’ and ‘Intensity of performance’. First, the child’s interest in paper-and-pencil tasks is evaluated. Then, the child is encouraged to complete a drawing booklet with seven paper-and-pencil tasks while a trained assessor observes and scores the performance and quality of the paper-and-pencil tasks. 

The subdomain ‘Task performance’ (i.e., WRITIC-TP) within the ‘Paper-and-pencil tasks’ domain was developed as a norm-referenced part of the WRITIC. The WRITIC-TP consists of seven items scoring the quality of paper-and-pencil tasks (tracing double-lined paths, coloring, making arcades, making garlands, name writing, making spirals, and copying letters and numbers) on a 3-point scale (range 0–2, maximum score: 14), and the performance of these tasks regarding sitting posture and pencil grip: five items (type of pencil grip, sitting posture, forearm position, distal versus proximal movement, and another hand) are scored on a 7-point scale with a range of 0 to 6 resulting in a maximum score of 30, and one item (wrist position) on a 5-point scale (range 0–4, maximum score: 4). Thus, the total score ranges from 0 to 48 [24].

Research on WRITIC-TP confirmed high internal consistency (α = 0.82), significant ability to discriminate between children with good and poor performance on paper-and-pencil tasks (U = 11.78, *p* < 0.001), and excellent test-retest and inter-rater reliability, with ICCs of 0.92 and 0.95, respectively [22]. WRITIC-TP, administered in kindergarten, is found to be the primary predictor for handwriting quality evaluated by the Systematic Screening for Handwriting Difficulties [23] in grade 1 [30]. 

### 2.3. Translation and Cultural Adaptation

WRITIC was translated following guidelines for translation and validation instruments in cross-cultural healthcare research [31,32,33].

After contacting the original authors and receiving their formal authorization and training, we moved on to translations. Two bilingual translators translated the instrument forward to the target language, followed by a consensus version performed by a third bilingual translator familiar with the instrument. An expert performed the back-translation to ensure that the correct meaning was obtained. Differences were discussed to get the final version of the instrument [29,34]. To clarify some inconsistencies in this latest translation, the document was analyzed by a group of experts in the field [35,36] until a total agreement was reached assuming conceptual meaning [30]. Finally, pilot testing was developed with six children in the age group for which the instrument is intended to test its application as suggested [32,37]. Thus, it was possible to verify a good percentage of understanding [38]. These children did not belong to the recruited sample.

### 2.4. Participants

A total of 88 children (47 boys and 41 girls; mean age 5.44, SD 0.50) from public/private kindergartens in the district of Évora (Portugal) were recruited during the first trimester of the school year. Most of the children (i.e., 81 or 92%) were right-handed, 5 (5.7%) were left-handed, and 2 (2.3%) were ambidextrous (Table 1). 

### 2.5. Data Collection

Data collection was conducted at three moments. For the first time, 70 children were assessed (data were used for the study on internal consistency and interrater reliability, although for the latter, one of the observers failed to observe one of the children, ending up being considered only 69). The second time, we retested 65 children from the previous 70 because five children missed kindergarten on that day (data were used for the test-retest reliability study). Finally, during the third time, in addition to the 70 children assessed the first time, an additional 18 were assessed. However, one was excluded for not completing all required tasks (convergent validity study).

WRITIC assessment took place in the classroom as suggested by the authors, with the child sitting on their chair at the desk and with other children in the classroom. 

### 2.6. Procedures 

#### 2.6.1. Content Validation

The selection of experts to review and discuss this instrument was based on individual expertise with the topic to be studied [39]. Six experts in handwriting (psychomotor and occupational therapists), who have professional experience working with preschool children, answered the questionnaire once.

Content validation procedure with WRITIC-PT was studied through a Delhi study [40]. 

We used a Likert scale scored between 1- and 4-points regarding consensus in relevance, coverage, and clarity of items, as was performed in the content validity study with the original instrument. The questionnaire was composed of thirty-one items related to the three domains of WRITIC (Table 2).

#### 2.6.2. Reliability

The main goal of reliability is to understand the consistency of the test scores [41]. We studied internal consistency, test-retest reliability, and inter-rater reliability to assess this psychometric property.

##### Internal Consistency

Internal consistency describes the extent to which the items in a test measure the same concept or construct. Internal consistency measures correlations between different items on the same test. Internal consistency was established with 70 children.

##### Test-Retest and Inter-Rater Reliability

To determine the test-retest reliability of WRITIC-PT, the device was administered, and after three weeks, it was re-administered under the same conditions in the same group of children. To evaluate inter-rater reliability, two different raters observed the same children simultaneously. 

Test-retest reliability was established with 65 children, and inter-rater reliability with 69 children. 

#### 2.6.3. Convergent Validity

Convergent validity assesses how two different instruments manage to collect the same information. Although convergent validity is not equivalent to construct validity, it is one piece of evidence used to judge the construct validity of the measure [42]. When possible, the collection of data to obtain the convergent validity should be carried out in the same time space of test-retest and inter-rater reliability. This study established convergent validity with 87 children. With this study, we used the same instruments on performance components as the authors of the WRITIC assessment used in the original convergent validity study by correlating WRITIC with Beery™VMI-6 and with 9-HPT with the dominant hand (as the original study).

The Beery™VMI-6 is commonly used to assess visual-motor integration [43] and gives essential information on the development of visual and motor abilities. In the visual motor integration test, children must copy a sequence of 21 images from a model (beginning with simple lines and progressing to complex geometric shapes). In the visual perception subtest, children must identify each item’s identical match from a set of similar figures. In the motor coordination subtest, children must trace the interior of the 21 geometric figures without crossing over the shape´s border [22]. Beery et al. [14] stated good psychometric properties, by age group, including test-retest reliability of 0.88 and interrater reliability of 0.93. In the psychometric study carried out with a Portuguese population, a good internal consistency index was found for the visual perception test (Cronbach’s alpha = 0.76) and the motor coordination test (Cronbach’s alpha = 0.72), being at the lower limit for the visual-motor integration test (Cronbach’s alpha = 0.68) [44].

The 9-HPT measures fine motor dexterity in terms of the number of seconds (i.e., completion time) the child has to place nine pegs in a pegboard and then remove them as quickly as possible [45]. The 9-HPT presents a high inter-rater and test-retest reliability (range, r = 0.86–0.98) [46], discriminates manual dexterity in multiple sclerosis patients from healthy controls at a highly significant level (*p* < 0.05) [47], and is frequently selected as a gold standard or reference value. It is also sensitive to treatment, which is the reason why it is recommended for inclusion in clinical trials [46].

### 2.7. Statistical Analysis

Data were analyzed using the Statistical Package for the Social Sciences, version 24.0, for Windows (SPSS Inc., Chicago, IL, USA). Tests of normality and descriptive statistics were performed for all outcome variables as a function of age and manual preference. Differences in the mean values between the groups for non-normally distributed measures were examined using the nonparametric Mann–Whitney U test. Statistical significance was set at *p* ≤ 0.05. 

#### 2.7.1. Content Validity

To determine content validity, we used a Content Validity Coefficient (CVC) to quantify and interpret the judgment of items [48].

#### 2.7.2. Reliability

Internal consistency is analyzed with the Cronbach alpha. The interpretation is based on acceptable alpha values ranging from 0.70 to 0.95 [49].

The Intraclass Correlation Coefficients (ICC) is widely used to calculate test-retest and inter-rater reliability. ICC values less than 0.5 indicate poor reliability, values between 0.5 and 0.75 indicate moderate reliability, and values between 0.75 and 0.9 indicate good reliability. Values greater than 0.90 indicate excellent reliability [50].

#### 2.7.3. Convergent Validity

To analyze the relationship among all variables, Spearman’s correlation coefficients were calculated. The strength of the relationship was set at r < 0.25, indicating a weak effect, r = 0.25–0.5 a moderate effect, r = 0.5–0.75 a strong effect, and r > 0.75 a very strong effect [51]. 

## 3. Results

The results by age on the sub domains task performance and intensity performance of the WRITIC are shown in Table 3 and by hand preference in Table 4:

### 3.1. Content Validation

The CVC for the entire instrument, considering the average for each of the three aspects judged—relevance, coverage, and clarity—was 0.93. 

Table 5 shows the CVC results for each item (Item 1–6: Child; Item 7: Sustained Attention; Item 8–9: Environment; Item 10–13: Pencil Grip; Item 14–24: Sitting Posture; Item 25–31: Paper-and-Pencil Tasks). 

In content validation, it was found that practically all items had scores above 0.8. Item 1 (How often do you color or do you make a drawing outside of school either at home or somewhere else?), item 3 (“How well do you think you are at drawing and coloring?”), item 6 (“How well do you think you are at ‘Writing’?”), and item 19 (shoulder position) had one aspect under 0.8 (0.79).

### 3.2. Reliability

#### 3.2.1. Internal Consistency

The sample of 70 children gave us the first evidence for internal consistency on the subdomain ‘Task Performance’ (Cronbach’s alpha of 0.72), the norm-referenced subdomain. Cronbach’s alpha indicates good internal consistency if it is higher than 0.7 [52].

On the other criterion-referenced subdomain, ‘Intensity of Performance’, we found a Cronbach’s alpha of 0.563, which means poor reliability. Through the analyses, we observed that if the item ‘Distance Nose-Desk’ were excluded, Cronbach´s alpha would be 0.717.

#### 3.2.2. Test-Retest and Inter-Rater Reliability

As can be seen in Table 6, test-retest reliability had good reliability on the norm-referenced subdomain ‘Task Performance’ with an Intra-class Correlation Coefficient (ICC) of 0.879 (*p* < 0.001). On the other criterion-referenced subdomains—‘Interest’, ‘Sustained Attention’, and ‘Intensity of Performance’—we found reliability with ICCs ranging from 0.652 to 0.719 (*p* < 0.001).

On the subdomains ‘Physical environment’ and ‘Social Environment’, we found an ICC of 1 (*p* < 0.001). The lack of changes in the environment over time justifies these results.

We found an excellent inter-rater reliability on the norm-referenced subdomain ‘Task Performance’ with an ICC of 0.925 (*p* < 0.001) and an excellent inter-rater reliability on the criterion-referenced subdomain ‘Intensity of Performance’ with an ICC of 0.829 (*p* < 0.001). 

### 3.3. Convergent Validity

We found statistically significant differences when we compared different ages. Six years-old children achieved more points on average on the norm-referenced subdomain ‘Task Performance’ and Intensity of Performance.’ When we compared results using the Mann–Whitney Test, we verified statistically significant differences in the norm-referenced subdomain ‘Task Performance’ (U = 691.000; *p* < 0.005) and in the quality of results in the criterion-referenced subdomain ‘Intensity of Performance’ (U = 700.500; *p* < 0.005). Children aged six years achieved more points on average in both tasks. 

We did not find statistically significant differences when we compared right-handed and left-handed children.

When we compared results using the Mann–Whitney Test, we did not verify statistically significant differences in the norm-referenced subdomain ‘Task Performance’ (U = 137.000; *p* > 0.005) and in the quality of results in the criterion-referenced subdomain ‘Intensity of Performance’ (U = 163.000; *p* > 0.005). 

This validity measures whether constructs that theoretically should be related to each other are related to each other. The correlations observed to establish convergent validity were moderate, with r ranging from 0.39 to 0.65 (Table 7).

## 4. Discussion

With the current research, which we did after developing the translation and cultural adaptation process of the WRITIC into European Portuguese, we determined the content validity, test-retest, and interrater reliability and the convergent validity with the Beery-VMI and 9-HPT on the norm-referenced subdomain of the Portuguese (PT) WRITIC Task Performance (TP). 

The project started with the translation and cultural adaptation of WRITIC, made by two bilingual translators, followed by our psychometric research.

Content validity was obtained after a high agreement amongst the expert panel. It became clear that the high consensus in the Portuguese version is in line with the original Dutch instrument [24].

Regarding internal consistency, we were able to verify a good Cronbach’s alpha for the norm-referenced subdomain ‘Task Performance’ and a poor Cronbach’s alpha for the criterion-referenced subdomain ‘Intensity of Performance’. Our scores were relatively lower when compared with the original study [24]. This may be since the sample size in our study was smaller than in the original study, considering that Cronbach’s alpha coefficient has been criticized by some authors, mainly because it is influenced by sample size and also by the instrument number of items [49,51]. Despite the relatively lower results of the Portuguese study, the internal consistency for the subdomain referenced to the norm in the literature [52] was satisfactory, which means good reliability. Still, it is essential to analyze the evolution of the Alpha as the sample increases.

On test-retest and inter-rater reliability, we conclude that the WRITIC-PT is stable over time and between raters. We obtained good results on the norm-referenced subdomain ‘Task Performance’, allowing us to see similar and positive results on the norm-referenced subdomain when compared with the original study [24].

On test-retest reliability, we observed that the WRITIC domain ‘Child’ had slightly lower results (the subdomain ‘Interest’ ICC is 0.67; the subdomain ‘Sustained Attention’ ICC is 0.65). This may have happened because the ‘Child’ domain involves answering opinion questions based on the child’s sustained attention and reaction to external stimuli, which can influence the child’s behavior differently on the two occasions and, consequently, our results. 

On inter-rater reliability, we obtained excellent results that sustained our study, despite having two evaluators. These results align with the ICCs of the original study [24]. 

On convergent validity, we found a good correlation between WRITIC-TP (Task Performance) and Beery^TM^VMI-6 and a moderate correlation with the 9-HPT. Compared with the original study, these results support the construct of handwriting readiness of WRITIC-PT [24].

In the future, we want to increase the sample in different parts of the country and apply it to children flagged as having weaknesses in handwriting skills.

## 5. Limitations and Future Directions

The current study needs to be interpreted considering both its strengths and limitations. Firstly, the stability of the sample size was difficult to maintain during the study. Furthermore, for future studies, we suggest recruiting a higher number of participants to be able to carry out more robust statistical procedures. 

Assessing children in their classroom environment gives researchers realistic context and data. On the other hand, this type of assessment can also cause some noise in the observations when the test is administered. Furthermore, WRITIC-PT should be performed over a digitized table to get additional temporal, spatial, and kinematic data to complement data and go through a highly consistent analysis.

Finally, we intend to continue the studies on WRITC-PT with typically developing children and children with atypical development.

## 6. Conclusions and Implications for Practice

Translation and cross-cultural adaptation of WRITIC into European Portuguese was a successful work conducted with the collaboration of authors in the Netherlands and the students and teachers at Évora University.

In this study, we established internal consistency on the subdomain ’Task Performance’ and obtained high to excellent test-retest and inter-rater reliability values. We determined convergent validity and found good (with Beery^TM^VMI-6) to moderate correlations (with 9-HPT) tests that underpin the construct proposed by WRITIC. This gives us confidence in using WRITIC-PT when assessing the handwriting readiness for the writing of children aged 5 and 6 years. The fact that these two tests are partly related to WRITIC-PT suggests that both instruments are valuable additional instruments to WRITIC-PT. In the case of non-handwriting readiness, according to WRITIC-PT, it is recommended to assess performance components in a second step with Beery^TM^VMI-6 and 9-HPT.

The Portuguese WRITIC is a valuable new assessment to evaluate handwriting readiness and gives a new understanding of the writing competence of Portuguese children before entering school. This instrument is valid and reliable and should be used to identify children who are not ready for handwriting. It assists in the goal of timely intervention for 5- and 6-year-old children and, thus, prevent handwriting difficulties in later grades.

This research provides a promising step toward quantifying handwriting readiness in Portuguese children, giving us the confidence to start the implementation process of WRITIC-PT in Portugal. 

## Figures and Tables

**Table 1 children-10-00559-t001:** Descriptive data of participants.

n = 88	Total Group	5 Years	6 Years
Number of participants	88 (100%)	49 (55.7%)	39 (44.3%)
Number of boys	47 (53.4%)	27 (30.7%)	20 (22.7%)
Number of girls	41 (46.6%)	22 (25%)	19 (21.6%)
Right-handed	81 (92%)	45 (51.1%)	36 (40.9%)
Left-Handed	5 (5.7%)	2 (2.3%)	3 (3.4%)
Variable-Handed	2 (2.3%)	2 (2.3%)	0 (0%)

**Table 2 children-10-00559-t002:** Identification of items by area of analysis in the questionnaire for content validation.

Child	Item 1–6	Child’s Interests
	Item 7	Sustained Attention
Environment	Item 8–9	Physical environment
	Item 9	Social environment
Paper-and-pencil tasks	Item 10–13	Pencil Grip
	Item 14–24	Sitting Posture
	Item 25–31	Quality of paper-and-pencil tasks

Note: CVC values accepted to consider the quality of an aspect, or an item judged should be greater than 0.80 [34].

**Table 3 children-10-00559-t003:** Task and intensity performance results by age.

n = 88	5 Years (n = 49)	6 Years (n = 39)
Task Performance	40.76 ± 3.35	42.21 ± 5.48
Intensity of Performance	20.67 ± 3.32	21.77 ± 3.33

**Table 4 children-10-00559-t004:** Task and intensity performance average results by hand preference.

n = 88	Right-Handed n = 81	Left-Handed n = 5	Ambidextrous n = 2
Task Performance	41.43 ± 4.37	43.80 ± 3.27	34 ± 3.00
Intensity of Performance	21.20 ± 3.36	22.60 ± 1.67	16 ± 0.00

**Table 5 children-10-00559-t005:** Content Validity Coefficient (CVC) results for each item in three aspects judged for Content Validation.

Item	Relevance	Coverage	Clarity
1	0.9	0.9	0.79 *
2	0.96	1.00	0.96
3	0.83	0.79 *	1.00
4	0.96	0.92	0.83
5	1.00	0.96	0.88
6	0.88	0.79 *	1.00
7	1.00	0.92	0.92
8	0.96	0.92	0.83
9	0.96	0.88	0.92
10	1.00	0.96	0.83
11	1.00	0.96	0.92
12	1.00	0.96	1.00
13	1.00	0.96	0.96
14	0.96	0.92	0.88
15	0.96	0.96	1.00
16	0.92	0.96	0.96
17	0.92	0.92	0.92
18	0.92	0.92	0.92
19	0.92	0.83	0.79 *
20	0.92	0.92	0.83
21	0.92	0.96	0.88
22	0.88	0.92	0.88
23	0.92	0.92	0.92
24	0.92	0.92	1.00
25	1.00	0.96	0.96
26	0.96	1.00	0.96
27	1.00	0.96	0.88
28	0.96	0.96	0.88
29	1.00	1.00	1.00
30	0.92	0.88	0.92
31	1.00	0.92	1.00

* Values that present CVC < 0.8 suggest that the item may present some type of distortion or problem in the judged aspect.

**Table 6 children-10-00559-t006:** Test-retest reliability ICC values.

n = 65	1st Evaluation	2nd Evaluation	ICC	99%IC
Task Performance	42.08 ± 4.37	41.51 ± 4.42	0.88	[0.77–0.94]
Intensity of Performance	21.77 ± 2.97	21.35 ± 2.77	0.72	[0.46–0.85]
Interest	8.91 ± 2.14	9.15 ± 1.96	0.67	[0.37–0.83]
Sustained Attention	1.89 ± 0.40	1.88 ± 0.41	0.65	[0.33–0.82]

**Table 7 children-10-00559-t007:** Results of correlations between two different instruments (Beery™VMI-6 and 9-HPT) and WRITIC.

n = 87 ^a^	Visual Motor Integration	Visual Perception	Motor Coordination	9-HPT
Task Performance	0.65 **	0.39 **	0.55 **	−0.40 **

** The correlation is significant at the 0.01 level. ^a^ One of 88 children was excluded.

## Data Availability

The data presented in this study are available on request from the corresponding author.

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
