# Peer review of "Translation, Cross-Cultural Adaptation, and Psychometric Properties of Writing Readiness Inventory Tool in Context (WRITIC)"

_children, 2023, doi:10.3390/children10030559_

Round 1
Reviewer 1 Report
This work is interesting, and fits with the scopes of the Children journal. The experiments and analyzed are well performed. However, the manuscript presents several limitating points which needs to be addressed.
- english is poor. Many incorrect sentences, grammar and syntaxic errors.
- the introduction is messy. All the information is present, but not well organized. Namely, the objectives are not clearly stated.
- the WRITIC is not clearly presented and introduced, while it is the central point of the paper. Please give a detailed description.
- the reasons for choosing to test the correlations of the WRITIC-PT with the Beery and the 9-HPT are not explained, neither in the introduction nor in the objectives. These latter tests are mentioned only in the materials and methods section (section 3.5. page 5), but without any rationale explaining their use. This point should be clarified in the introduction and in the objectives.
- the objectives of the article are not clear enough. This part should be more detailed.
- ethics approval : in section 3.3 page 3, it is mentioned that the written informed consent was obtained from the participants. However the children in this study were not able to read and write. They only gave oral consent. What about the childrens’ parents ? Did the authors obtain written consent from all parents ?
- section 3.4. Reliability : no demographical information is given about participants (children) : number of participants (70 ? 87 ? 88 ?), mean age, sex ratio, handedness
- section 3.4.2. Participants : why was 1 of the 70 children not assessed by 2 different observers for inter-rater reliability ? The reason should be explained.
- page 6, line 232 : who are « MS » patients ? (explain the abbreviation)
- abbreviations are missing
- page 9 : why is demographical information about participants given only for convergent validity (table 4) ? This information must be given for all tests. Moreover, mean age should be provided.
- convergent validity : in the 3.5 « participants » section (page 6 line 205), it is written that this validity was performed with 87 children. But then, in the 4.3. section of the Results, 88 participants are mentioned (Table 4, page 9). It is then precised in the legend of Table 7 (page 10) that n=87 because one of the 88 children was excluded, but without any explanation. All this must be explained in details in the Materials & Methods section. I would recommend a unique section in the Materials and Methods section presenting all participants, with a unique Table with the demographic data for all groups of participants depending on the test.
- discussion is too weak. The main results are presented again, but only slightly discussed in regards to the litterature. The discussion needs to be more thorough.
- limitations of the study are only briefly discussed. A limitations section may be added, before the conclusion.
Author Response
Dear Reviewer,
We thank you for reviewing our paper entitled Translation, cross-cultural adaptation, and psychometric properties of Writing Readiness Inventory Tool in Context (WRITIC).
We appreciate the insightful comments that allowed us to improve the paper further and thank the editors for the opportunity to resubmit the manuscript.
Your comments have been addressed. The changes have been made in the revised paper and are marked with the track changes option. These changes it is included a thorough US English spelling and grammar checker.
As requested, we have added some new references (improving our introduction and conclusions).
We hope that our revisions are satisfactory and that the article can now be accepted for publication in Children’s Special Issue: Child Psychomotricity: Development, Assessment, and Intervention. We look forward to hearing from you.
Comments and Suggestions for Authors
Comment 1 (C1): “The introduction is messy. All the information is present, but not well organized. Namely, the objectives are not clearly stated”.
Answer 1 (ANS1): Thank for your suggestion, we have added information and references and relocated some topics in the introduction (Page 1: line 35 - page 2: line 73). We substantiated the objective of the study, as stated in the last sentence of the introduction (page 2: line 106-108).
C2: “The WRITIC is not clearly presented and introduced, while it is the central point of the paper. Please give a detailed description”
ANS2: Thanks for your suggestion. We have added a detailed description of WRITIC assessment (Page 4: line 120 – 143).
C3: “the reasons for choosing to test the correlations of the WRITIC-PT with the Beery and the 9-HPT are not explained, neither in the introduction nor in the objectives. These latter tests are mentioned only in the materials and methods section (section 3.5. page 5), but without any rationale explaining their use. This point should be clarified in the introduction and in the objectives.”
ANS3: Thank you for your comment, we have added information on the performance components for handwriting and the corresponding tests; Beery VMI and 9-HPT (Page 2: line 58-62). This information has been reinforced in Methods section (Page 6-7: line 233 - 245)
C4: the objectives of the article are not clear enough. This part should be more detailed.
ANS4: Thank you for your comment, we described the objective of this study in more detail (Page 3: line 146-148)
C5: Ethics approval : in section 3.3 page 3, it is mentioned that the written informed consent was obtained from the participants. However the children in this study were not able to read and write. They only gave oral consent. What about the childrens’ parents? Did the authors obtain written consent from all parents ?
ANS5: Thank you for your comment, we added information on informed consent from the parents and provided assent from the children (Page 3: line 106-108).
C6: section 3.4. Reliability : no demographical information is given about participants (children) : number of participants (70 ? 87 ? 88 ?), mean age, sex ratio, handedness
ANS6: Thank you for your comment, we have added this information (Page 5: line 165-171).
C7: section 3.4.2. Participants : why was 1 of the 70 children not assessed by 2 different observers for inter-rater reliability ? The reason should be explained.
ANS7: Thank you for your comment, we have added more detailed information regarding the participants (Page 6: line 210-221).
C8: page 6, line 232 : who are « MS » patients ? (explain the abbreviation)
ANS8: Thank you for the comment we have added the full name of the disease instead of the abbreviation (Page 7: line 249.)
C9: page 9 : why is demographical information about participants given only for convergent validity (table 4) ? This information must be given for all tests. Moreover, mean age should be provided.
ANS9: Thank you for the suggestion, we added this information for all test (Page 5: line 165-168).
C10. convergent validity : in the 3.5 « participants » section (page 6 line 205), it is written that this validity was performed with 87 children. But then, in the 4.3. section of the Results, 88 participants are mentioned (Table 4, page 9). It is then precised in the legend of Table 7 (page 10) that n=87 because one of the 88 children was excluded, but without any explanation. All this must be explained in details in the Materials & Methods section. I would recommend a unique section in the Materials and Methods section presenting all participants, with a unique Table with the demographic data for all groups of participants depending on the test.
ANS10: Thank you for your comment, we have added the requested information throughout the Materials & Methods section.
C11: discussion is too weak. The main results are presented again, but only slightly discussed in regards to the literature. The discussion needs to be more thorough.
ANS11: Thank you for your comment. We have updated the discussion with literature and also added a limitation section. The aim of this study is to establish the reliability and validity of WRITIC-PT and to relate the outcomes to the outcomes of the original Dutch WRITIC. Therefore, this is central to the discussion
C12: limitations of the study are only briefly discussed. A limitations section may be added, before the conclusions
ANS12: See ANS11.
Reviewer 2 Report
Manuscript ID: Children-2156577
Full Title: Translation, cross-cultural adaptation, and psychometric properties of Writing Readiness Inventory Tool in Context (WRITIC)
The article describes the translation and cultural adaptation process of the WRITIC (Writing Readiness Inventory Tool in Context) into European Portuguese.
However, in reviewing the paper, I noticed some issues that should be addressed before this review might be considered for publication.
The main concern is regarding the Methods section:
· The significant necessary information of how the sample size was calculated and according to which quality criteria (for example, the COSMIN group: Available online: https://www.cosmin.nl) is missing. Also, although it seems that the sample collected for all phases contained the same children (except the additional 17 participants in the convergent validity assessment phase), the authors described participants and procedure for each phase separately. the repeated description is redundant, making the article unnecessarily long. Furthermore, the separate descriptions incorrectly implied different children in each phase, and so the actual sample(s) are not clearly described. Further, the description is inconsistent: inclusion and exclusion criteria were described for the reliability phase (lines 158-163) but not for any other phase.
· The Methods section would benefit from reorganization. The Study Design and Participants information could be collected into one paragraph, and then all necessary missing information could be added.
· The statistical analysis should be described in a separate section with detailed explanations of the tests used for each psychometric property assessment.
General comment:
The manuscript should be edited to improve fluency and language.
More suggestions should be considered:
Introduction
· The existing research literature discerns between the two key terms: handwriting and writing, but these terms are used in the manuscript interchangeably with no explanations about the differences or relationships between them. For example, the authors choose “handwriting readiness” as a keyword, although the tool name is “Writing Readiness Inventory Tool in Context.” The first line of the Introduction (line 33) talks about the importance of writing, but the second line (line 34) refers to handwriting. This issue should be checked throughout the manuscript, and the relationships between the terms should be explained to readers very clearly.
· I suggest the authors update the Introduction to include new evidence published in recent years that might enrich the background of their research (e.g., https://doi.org/10.1080/01942638.2021.1912247). Also, in continuing the description of the connections between writing and reading, I suggest adding references that describe evidence for different and similar patterns in handwriting and drawing (e.g., https://doi.org/10.3389/fpsyg.2020.01810).
· Because there are significant differences between countries, a short explanation should be added to the Introduction regarding what is known and unknown about participation in handwriting activities in kindergartens around Portugal. Is there any national program? How many hours do children spends on handwriting tasks? What kind of tasks are usually done in Portuguese kindergartens?
Methods
· There is no mention of when the study was conducted (the specific period of data collection).
· It is not clear to which part of the manuscript paragraph 2, “WRITIC-PT,” belongs. Is it part of the Introduction?
· I suggest moving the tool description to the Methods section.
· It would have helped to start the Methods section with a short, clear description of the different phases of the study (e.g., first, the original WRITIC-TP was cross-culturally adapted from English to Portuguese; then the psychometric properties of the Portuguese version of WRITIC-TP were assessed for (a) content validity, (b) reliability, etc.)
· Paragraph 3.1. Ethics approval can be added to that opening description or under the study design section.
· Right after that short description of the phases, I suggest adding the paragraph describing the tool (now, paragraph 2). Include details regarding grades and some examples of the tasks.
· “Finally, pilot testing was developed with children in the age group” (lines 123–124). What was the sample size in this pilot?
· Were inter-subscale correlations evaluated? If not, the authors should consider doing so or describing the rationale to not do so.
Results
· The descriptive data of participants should be the first paragraph in the Results section.
· Because the current description of the WRITIC-TP tool did not contain any description of the grades expected in each subscale, it is difficult to understand the meaning of some of the presented results (e.g., without that information, I cannot understand the scores of Evaluations 1and 2 presented in Table 3, line 273 or the task and intensity performance presented in Table 5, line 296).
Discussion
· Repeated reporting of the results takes a large volume from the Discussion section—unnecessarily adding length without adding content. I suggest you fully report the results in the Results section (as per my comment above) and use the Discussion section to add to and deepen your explanation of those results. That would contribute insights for others who are considering using this tool.
· Add a section for the study limitations.
· “After some time when two bilingual translators made a translation …” (line 329) –Please be specific how much time is “some time”.
· The proposed explanation for lower results on test–retest reliability needs to be expanded. What’s the difference between the child domain and the others?
· Please correct: Replace “a” with “A” in the sentence: “In the future we want to increase the sample in different parts of the country. As well as apply it to children who are flagged as having weaknesses in writing skills” (lines 369-370).
· Please correct: Replace “s” with “S” in the sentence: “The Portuguese version has good psychometric properties. Similar to the original version.” (lines 388-389)
References
Many references lack a DOI address.
Author Response
Dear Reviewer,
We thank you for reviewing our paper entitled Translation, cross-cultural adaptation, and psychometric properties of Writing Readiness Inventory Tool in Context (WRITIC).
We appreciate the insightful comments that allowed us to improve the paper further and thank the editors for the opportunity to resubmit the manuscript.
Your comments have been addressed. The changes have been made in the revised paper and are marked with the track changes option. These changes it is included a thorough US English spelling and grammar checker.
As requested, we have added some new references (improving our introduction and conclusions).
We hope that revisions are satisfactory and that the article can now be accepted for publication in Children’s Special Issue: Child Psychomotricity: Development, Assessment, and Intervention. We look forward to hearing from you.
Comments and Suggestions for Authors
Introduction
Comment 1 (C1): the existing research literature discerns between the two key terms: handwriting and writing, but these terms are used in the manuscript interchangeably with no explanations about the differences or relationships between them. For example, the authors choose “handwriting readiness” as a keyword, although the tool name is “Writing Readiness Inventory Tool in Context.” The first line of the Introduction (line 33) talks about the importance of writing, but the second line (line 34) refers to handwriting. This issue should be checked throughout the manuscript, and the relationships between the terms should be explained to readers very clearly.
Answer 1 (ANS1): Thank you for your comment, throughout the manuscript, the term handwriting was standardized. When developing the instrument, the authors chose this name for the instrument because of the acronym that refers to (hand)writing.
C2: I suggest the authors update the Introduction to include new evidence published in recent years that might enrich the background of their research (e.g., https://doi.org/10.1080/01942638.2021.1912247). Also, in continuing the description of the connections between writing and reading, I suggest adding references that describe evidence for different and similar patterns in handwriting and drawing (e.g., https://doi.org/10.3389/fpsyg.2020.01810).
ANS2: Thank you for your suggestions regarding the introduction and thank you for the referenced literature. We have updated the introduction and have added the cited literature (Page 1: line 35 – page 2: lines 46 and 55).
C3: Because there are significant differences between countries, a short explanation should be added to the Introduction regarding what is known and unknown about participation in handwriting activities in kindergartens around Portugal. Is there any national program? How many hours do children spends on handwriting tasks? What kind of tasks are usually done in Portuguese kindergartens?
ANS3: Thank you for your suggestion, we have added information about handwriting activities and education in the Portuguese context (page 2: line 63-70).
Methods section
C4: The significant necessary information of how the sample size was calculated and according to which quality criteria (for example, the COSMIN group: Available online: https://www.cosmin.nl) is missing. Also, although it seems that the sample collected for all phases contained the same children (except the additional 17 participants in the convergent validity assessment phase), the authors described participants and procedure for each phase separately. the repeated description is redundant, making the article unnecessarily long. Furthermore, the separate descriptions incorrectly implied different children in each phase, and so the actual sample(s) are not clearly described. Further, the description is inconsistent: inclusion and exclusion criteria were described for the reliability phase (lines 158-163) but not for any other phase.
The Methods section would benefit from reorganization. The Study Design and Participants could be collected into one paragraph, and then all necessary missing information could be added.
ANS4: Thank you for your suggestion, we have added more detailed information regarding the participants (Page 6: line 164).
C5: The statistical analysis should be described in a separate section with detailed explanations of the tests used for each psychometric property assessment.
ANS5: Thank you for your comment, we have integrated the suggested section (page 7: line 254).
C6: There is no mention of when the study was conducted (the specific period of data collection).
ANS6: Thank you for your comment, we have added this information (Page 5: line 166-167).
C7: It is not clear to which part of the manuscript paragraph 2, “WRITIC-PT,” belongs. Is it part of the Introduction?
ANS7: Thank you for your comment, this information was clarified on page 3, line 112.
C8: I suggest moving the tool description to the Methods section.
ANS8: Thank you for your suggestion, we have moved the tool description to the Material and Methods section.
C9: It would have helped to start the Methods section with a short, clear description of the different phases of the study (e.g., first, the original WRITIC-TP was cross-culturally adapted from English to Portuguese; then the psychometric properties of the Portuguese version of WRITIC-TP were assessed for (a) content validity, (b) reliability, etc.)
ANS9: Thank you for your comment, we have added this description (Page 3: line 110-113).
C10: Paragraph 3.1. Ethics approval can be added to that opening description or under the study design section.
ANS10: Thank you for your comment, we have added this information (Page 3: line 115-118).
C11: Right after that short description of the phases, I suggest adding the paragraph describing the tool (now, paragraph 2). Include details regarding grades and some examples of the tasks.
ANS11: See ANS8
C12: “Finally, pilot testing was developed with children in the age group” (lines 123–124). What was the sample size in this pilot?
ANS12: Thank you for your comment, we have added this information (Page 3: line 111) .
C13: Were inter-subscale correlations evaluated? If not, the authors should consider doing so or describing the rationale to not do so.
ANS13: Thank you for your comment. We did not evaluate inter-subscale correlations. Our study is a replica of the Dutch psychometric study of the original instrument. Since the original study did not correlate between subscales, we also did not.
Results
C14: The descriptive data of participants should be the first paragraph in the Results section.
ANS14: Thank you for your comment, we have relocated this information (Page 5: line 165-171).
C15: Because the current description of the WRITIC-TP tool did not contain any description of the grades expected in each subscale, it is difficult to understand the meaning of some of the presented results (e.g., without that information, I cannot understand the scores of Evaluations 1and 2 presented in Table 3, line 273 or the task and intensity performance presented in Table 5, line 296).
ANS15: Thank you for your comment. We have added Information regarding the range and maximum scores of the sub domain Task Performance of the instrument (Page 4: line 134-143).
Discussion
C16: Repeated reporting of the results takes a large volume from the Discussion section—unnecessarily adding length without adding content. I suggest you fully report the results in the Results section (as per my comment above) and use the Discussion section to add to and deepen your explanation of those results. That would contribute insights for others who are considering using this tool.
ANS16: Thank you for the suggestion. In addition to the results described in the Result section, we have included the interpretation of the results plus the similarities and differences of the outcomes with the outcomes of the original study of the instrument and the discussion with other literature in the Discussion section.
C17: Add a section for the study limitations.
ANS17: Thank you for your suggestion. We have added a section on study limitations (Page 12: line 408-419).
C18: “After some time when two bilingual translators made a translation …” (line 329) –Please be specific how much time is “some time”.
ANS18: Thank you for your comment, we have rewritten this sentence (Page 4: line 154).
C19: The proposed explanation for lower results on test–retest reliability needs to be expanded. What’s the difference between the child domain and the others?
ANS19: Thank you for your comment. We have added some information (Page 11: line 379-392).
C20: Please correct: Replace “a” with “A” in the sentence: “In the future we want to increase the sample in different parts of the country. As well as apply it to children who are flagged as having weaknesses in writing skills” (lines 369-370).
ANS20: Thank you for your comment, the sentence has been corrected.
C21: Please correct: Replace “s” with “S” in the sentence: “The Portuguese version has good psychometric properties. Similar to the original version.” (lines 388-389)
ANS 21: Thank you for your comment, the sentence has been corrected.
References
C22: Many references lack a DOI address.
ANS22: Thank you for your comment, we have been added the DOI’s in the references.
Round 2
Reviewer 1 Report
Thank you to the authors who carefully enhance their manuscript according to our comments. I have only a few, very minor, corrections to add:
- lines 49 to 52 page 2 : the word « handwriting » is repeated 4 times in 4 lines. Slightly modify this to make the reading more fluent.
- line 52 page 2 : replace « less space is left » by « less resources are available »
- line 79 page 2 : why is « not » between parenthesis ? remove the parenthesis
- line 87 page 3 : the data concerning the prevalence are not fully correct. Depending on the country and the method used by the authors, the proportion of children with handwriting difficulties varies between 5 and 34%. However the proportion of children with a clear, diagnosed dysgraphia, is only 5 to 10%. Please modify this sentence, and add references.
- line 111 page 3 : « translation and cross-cultural adaptation ARE PROCESSES that INVOLVE … »
- line 134 page 4 : remove from « WRITIC is administered… » to « paper-and-pencil tasks » (redundancy).
- lines 429 and 434 page 13 : replace « children with six years » by « Six years-old children »
- line 528 page 15 : replace « cinematic » by « kinematic »
Author Response
Dear Reviewer,
We want to thank you for your contributions to improving our manuscript.
Kind regards,
The Authors
Reviewer 2 Report
The manuscript was improved, and comments have been addressed.
Author Response

(The authors gave the same response as above.)
